# Pheochromocytoma–Paraganglioma Syndrome: A Multiform Disease with Different Genotype and Phenotype Features

**DOI:** 10.3390/biomedicines12102385

**Published:** 2024-10-18

**Authors:** Mara Giacché, Maria Chiara Tacchetti, Claudia Agabiti-Rosei, Francesco Torlone, Francesco Bandera, Claudia Izzi, Enrico Agabiti-Rosei

**Affiliations:** 1Department of Medical and Surgical Specialties, Division of Clinical Genetics, ASST-Spedali Civili, 25133 Brescia, Italy; 2Division Internal Medicine 2, Department of Clinical and Experimental Sciences, University of Brescia, ASST-Spedali Civili, 25133 Brescia, Italy; chiara.tacchetti@asst-spedalicivili.it (M.C.T.); claudia.agabitirosei@unibs.it (C.A.-R.); 3Clinical Research Hospital, IRCCS Multimedica, Sesto San Giovanni, 20099 Milan, Italy; francesco.torlone@multimedica.it (F.T.); francesco.bandera@multimedica.it (F.B.); 4Department of Medical and Surgical Specialties and Nephrology Unit, University of Brescia, ASST-Spedali Civili, 25133 Brescia, Italy; claudia.izzi@unibs.it; 5Department of Clinical and Experimental Sciences, University of Brescia, Clinical Research Hospital, IRCCS Multimedica, Sesto San Giovanni, 20099 Milan, Italy

**Keywords:** paraganglioma, pheochromocytoma, genetics, NF1, VHL, RET, SDHX, TMEM, MAX, FH

## Abstract

Pheochromocytoma and paraganglioma (PPGL) are rare tumors derived from the adrenal medulla and extra-adrenal chromaffin cells. Diagnosis is often challenging due to the great variability in clinical presentation; the complexity of management due to the dangerous effects of catecholamine excess and the potentially malignant behavior require in-depth knowledge of the pathology and multidisciplinary management. Nowadays, diagnostic ability has certainly improved and guidelines and consensus documents for treatment and follow-up are available. A major impulse to the development of this knowledge has come from the new findings on the genetic and molecular characteristics of PPGLs. Germline mutation in susceptibility genes is detected in 40% of subjects, with a mutation frequency of 10–12% also in patients with sporadic presentation and genetic testing should be incorporated within clinical care. PPGL susceptibility genes include “old genes” associated with Neurofibromatosis type 1 (NF1 gene), Von Hippel Lindau syndrome (VHL gene) and Multiple Endocrine Neoplasia type 2 syndrome (RET gene), the family of SDHx genes (SDHA, SDHB, SDHC, SDHD, SDHAF2), and genes less frequently involved such as TMEM, MAX, and FH. Each gene has a different risk of relapse, malignancy, and other organ involvement; for mutation carriers, affected or asymptomatic, it is possible to define a tailored long-life surveillance program according to the gene involved. In addition, molecular characterization of the tumor has allowed the identification of somatic mutations in other driver genes, bringing to 70% the PPGLs for which we know the mechanisms of tumorigenesis. This has expanded the catalog of tumor driver genes, which are identifiable in up to 70% of patients Integrated genomic and transcriptomic data over the last 10 years have revealed three distinct major molecular signatures, triggered by pathogenic variants in susceptibility genes and characterized by the activation of a specific oncogenic signaling: the pseudo hypoxic, the kinase, and the Wnt signaling pathways. These molecular clusters show a different biochemical phenotype and clinical behavior; they may also represent the prerequisite for implementing customized therapy and follow-up.

## 1. Introduction

Pheochromocytoma (PHEO) and paraganglioma (PGL) are rare neuroendocrine tumors: PHEOs arise from adrenal chromaffin cells (80–85%), while PGLs arise from extra adrenal chromaffin cells (15–20%) in parasympathetic and sympathetic ganglia located from the skull base, along the paravertebral region up to the pelvic floor. Parasympathetic paragangliomas are usually functionally silent, whereas sympathetic paragangliomas and pheochromocytomas generally produce catecholamines. In this review, we will focus on catecholamine secreting pheochromocytoma–paragangliomas (PPGLs).

The ability to produce catecholamines, sometimes responsible for dramatic clinical situations, the extreme variability of clinical presentation, and the resulting diagnostic challenge, together with the complexity of management, have made PPGL a fascinating disease, shrouded in an aura of mystery. However, in the last twenty years, progressive comprehension of genetics and molecular characteristics has allowed a better understanding of clinical behavior, prognosis, and therapy.

PPGL prevalence varies from 0.2 to 0.6% in hypertensive patients, to less than 0.05% in the general population. Although diagnostic delays are common, nowadays, the ability to diagnose the disease has certainly improved; actually, more than 60% of diagnoses rely on incidental findings, only one-third of cases are diagnosed as a consequence of PPGL-related symptoms, and about 10% thanks to the application of surveillance programs in patients carrying mutations in susceptibility genes [1]. 

## 2. Clinical Presentation

In this new context, diagnostic attitudes and sensibilities have gradually adapted; signs and symptoms associated with catecholamine hypersecretion are numerous and often non-specific. In fact, recently, a very interesting study by Geroula et al. [2] shed light on the symptoms that are significantly associated with the likelihood of a diagnosis of PPGL in a large cohort of subjects screened for suspected PPGL. Only hyperhidrosis, palpitations, pallor, tremor, and nausea were significantly associated with PPGL diagnosis; hypertension, headache, panic/anxiety, and flushing were not more frequent in PPGL patients compared with non-PPGL subjects, even if for a long time they have been considered pathognomonic for catecholamine hyper incretion. Interestingly, new-onset diabetes in young subjects was identified as a possible diagnostic clue and has been proposed in a document of a Working Group the European Society of Hypertension as a suspicion criterion for the biochemical screening of PPGL [3]. Frequency of symptoms and signs and their sensitivity for the diagnosis of PPGL are reported in Table 1. In addition, clinical conditions requiring screening for PPGL are reported in Table 2.

More attention is required for the biochemical screening not only of adrenal incidentalomas, which in 7% of cases turn out to be pheochromocytomas, but also for retroperitoneal masses. However, the appearance of episodic symptoms, the so-called “spells”, always constitutes a valid diagnostic clue: some patients experience symptoms in a paroxysmal manner, sometimes associated with an increase in blood pressure values; the frequency of attacks is extremely variable, ranging from a few attacks over several years to more than one per day. The attacks may occur spontaneously or may be triggered by an increase in abdominal pressure, alcohol, or medications. In this regard, hypertensive crises or hemodynamic instability induced by anesthesia should not be underestimated and require prompt biochemical screening. Orthostatic hypotension in untreated hypertensive subjects should also be regarded with suspicion, and although most subjects are asymptomatic, some patients suffer from dizziness and faintness; orthostatic hypotension may result from adrenergic receptor desensitization and intravascular volume depletion.

The excess of catecholamines is responsible for many effects on the cardiovascular system, in addition to hypertension, whether stable or paroxysmal. In fact, some patients may experience arrhythmias, myocardial ischemia, myocarditis, or cardiomyopathy, and a cross-sectional study detected an incidence of cardiovascular events in 19.3% of PPGL patients [5]. A recent study showed cardiovascular complications in 28% of patients with a difference according to the biochemical phenotype: patients with adrenergic phenotype more frequently had complications such as acute myocardial infarction or acute heart failure with a normal coronary angiogram, while patients with noradrenaline secretion had complications associated with atherosclerosis [6]. Catecholamine-induced cardiomyopathy should also be considered more carefully, since it complicates the clinical course of approximately 8–10% of PPGL cases; a recent meta-analysis by Cuspidi et al. [7] suggests that very early abnormality in LV systolic function, which is not detectable by conventional echocardiography in PPGL patients, may be detected by speckle tracking echocardiography, which investigates left ventricular mechanics via the global longitudinal strain, a more sensitive index of left ventricular function compared to ejection fraction. This means that cardiac involvement is probably much more frequent than previously believed. Although Takotsubo cardiomyopathy is the most popular form of catecholamine-induced cardiomyopathy, patients may develop dilated or hypertrophic cardiomyopathy [8]. In a recent state-of-the-art-review, the early cardiac involvement of the heart, as assessed by ECG and clinical imaging techniques, could be characterized as follows: (1) sinus tachycardia and arrhythmias, LVH, and repolarization abnormalities at ECG n; (2) LVH, decreased systolic strain, and decreased ejection fraction at echocardiography; (3) decreased systolic and diastolic strain, fibrosis and edema, and decreased ejection fraction at cardiac MR. These abnormalities may predispose to acute and chronic cardiac complications [9]. Regardless of cardiac remodeling, clinical presentation is usually dramatic with acute heart failure or acute coronary syndrome; often, the increase in myocardial cytonecrosis enzymes is modest and the coronary angiography is negative. PPGLs should always be excluded in subjects with acute coronary syndrome and a coronary angiogram not consistent with the clinical presentation. Patients with tumors releasing large amounts of catecholamines may develop a rare dreadful complication, i.e., the catecholamine crisis or phaeochromocytoma multisystem crisis, which consists of a severe hemodynamic instability that evolves to collapse and to irreversible shock. These patients, before developing hemodynamic impairment, despite very high levels of circulating catecholamines, may have low blood pressure values; prompt introduction of alpha-blocker therapy and fluid replacement are necessary to prevent progression to shock. For all that has been considered so far, whenever PPGLS is suspected, biochemical tests should be promptly carried out, regardless of whether the patient is hypertensive or not, and a direct imaging study should be favored in life-threatening situations.

## 3. Biochemical Diagnosis

Biochemical screening is the first diagnostic step and is based on the measurement of plasma or urinary free metanephrines; metanephrines are an expression of the intra-tumor metabolism of catecholamines and have a higher sensitivity and specificity than catecholamines, which depend on paroxysmal secretion and are influenced by sympathetic adrenergic activity. Both plasma free and urinary fractionated metanephrines have a very high sensitivity, of 98 and 93%, respectively [10], with negative predictive values greater than 99% [3], and therefore values in the normal range allow the diagnosis to be ruled out with reasonable certainty. Only very small tumors (<1 cm), which release a little amount of hormones, may escape diagnosis, but these are generally asymptomatic patients who are screened because they carry susceptibility gene mutations or because they have incidentalomas and can therefore be followed up with a surveillance program. More commonly, it is necessary to address the problem of false positives, considering that the specificity of plasma and urinary metanephrines is about 94% [10]; pharmacological interference is mostly due to tricyclic antidepressants [11], but other commonly used drugs (α-blockers, levodopa, MAO inhibitors, sympathomimetics, sulfasalazine) or certain food may increase catecholamine metabolites (Table 3).

Several diseases are associated with chronically elevated sympathetic activity, including heart failure, obstructive sleep apnoea, and renal failure but common false positives also derive from procedural error in sample collection for plasma metanephrines. The correct practice requires the patient to be in a supine position, for 30 min, under calm and quiet conditions before blood sampling; if the plasma sample is not correctly collectable, it is better to rely on urinary metanephrines [3]. Suspicion of PPGL may result from a two-fold increase in upper cut-off values according to the laboratory interval reference in one metabolite or any increase in two or more metabolites (high positive predictive values); in these cases, diagnostic imaging investigation is recommended.

## 4. Genetics and Molecular Biology

PPGLs are the most frequently inherited neoplasms: approximately 30–35% of patients have germline mutations in predisposition genes; in addition, somatic mutations are identified in 35–40% of sporadic PPGLS. This has expanded the catalog of tumor driver genes, which are identifiable in up to 70% of patients [12]. Integrated genomic and transcriptomic data over the last 10 years have revealed three distinct major molecular signatures, triggered by pathogenic variants in susceptibility genes and characterized by the activation of a specific oncogenic signaling: the pseudo hypoxic, the kinase, and the Wnt signaling pathways. These molecular clusters show a different biochemical phenotype and clinical behavior; they may also represent the prerequisite for implementing customized therapy and follow-up [12].

PPGLs with germinal/somatic mutations in the Krebs cycle-associated genes (SDHx, FH, MDH2, GOT2, SLC25A11, DLST, IDH1) and VHL/EPAS1-related genes (PHD1/2, EGLN1, HIF2a, EPAS1, IRP1) are included in the “pseudo-hypoxic cluster”, characterized by over-activation of the hypoxic pathway. These tumors are frequently extra-adrenal and commonly have a noradrenergic phenotype [13]. Cluster-1 chromaffin cells are less differentiated, lacking the last enzymatic step to produce adrenaline and are characterized by a constitutive secretion, so tumors have relatively catecholamine contents compared to adrenergic tumors. Cluster-1 patients have more frequently sustained hypertension rather than paroxysmal symptomatic hypertension and more frequently may suffer cardiovascular complications from arterial hypertension, diabetes, and increased arterial stiffness [5]. Due to these clinical features, cluster-1 tumors are more frequently asymptomatic and especially SDHB-PGLs may reach a large tumor size at diagnosis. The most relevant characteristic of cluster-1 PPGLs is that they may have biologically aggressive behavior and most malignant PPGLS belong to this cluster [14]. Some cluster-1 tumors may also be characterized by high dopamine secretion relative to normetanephrine, so they have a “dopaminergic phenotype”, determined on high plasma 3-methoxytyramine values compared to normetanephrine, as urinary dopamine is derived from the renal metabolism and excretion of circulating L-dopa. Plasma 3-methoxytyramine could also be a useful biomarker to assess the risk of malignancy and the expression of very-low differentiated chromaffin cells, which have lost enzymes to proceed in catecholamine synthesis.

The kinase pathway cluster includes tumors with germline/somatic mutations in NF1, RET, TMEM127, MAX, MET, MERTK, HRAS, FGFR1, and B-RAF. Molecular signatures rely on the over-activation of phosphatidylinositol-3-kinase (PI3K)/AKT, mTORC, and RAS/RAF/ERK signaling. In contrast with cluster-1, these tumors prefer an adrenal location, secrete adrenaline, and have paroxysmal symptoms. They usually have a high storage of catecholamines, due to the low rate of constitutive secretion, while the release is episodic and associated with accessional symptoms. Cluster-2 tumors more rarely, compared to cluster-1, have malignant behavior. Acute cardiovascular complications, such as Takotsubo-like cardiomyopathy, heart rhythm disorders, and hypertensive crisis are more frequent in cluster-1 patients [5].

The Wnt signaling pathway cluster is less characterized; no germline mutations have been identified that impact on Wnt signaling, only somatic driver fusion mutation involving the “mastermind-like” transcriptional coactivator 3 (MAML3) gene and somatic mutation in the cold shock domain-containing E1 (CSDE1) gene have been identified as driver molecular events in PHEO/PGL carcinogenesis. Activation of Wnt-signaling and bet-catenin constitute a possible target for tumors belonging to cluster-3.

An interesting meta-analysis by Crona et al. [15] on 703 patient from 21 studies, aimed at evaluating if driver mutations are valuable in predicting survival and malignant disease, classified patients according to different molecular subgroups according to SDHB mutation, attribution to cluster-1 or 2 or 3, and to four subgroups (tricarboxylic acid cycle, VHL/EPAS1, kinase signaling, or Wnt/unknown); the meta-analysis concluded that in a multivariate model, only the SDHB mutation, norepinephrine, and dopamine phenotype are correlated with the risk of metastasis, while no association is found for other molecular systems.

## 5. Inherited PPGL

While genetic analysis on tumor tissue is actually performed postoperatively and is reserved for patients with metastatic and inoperable disease with the goal of identifying a personalized therapeutic option, screening on germinal DNA has now been applied in clinical practice for 20 years; a genetic study enriches the diagnostic work-up and also adds useful information for a better characterisation of the disease in the pre-surgery phase. The presence of multiple PGLs, bilateral PHEO, and young age of onset (younger than 45 years) are suggestive of hereditary disease; however, the traces of heredity are often concealed. As in most hereditary neoplastic syndromes, predisposition to PPGL is transmitted in autosomal dominant fashion, but disease penetrance is often incomplete and subjects carrying pathogenic variants in susceptibility genes do not always develop features of the genetic disorder. So, up to 10–12% of sporadic and non-syndromic PPGLs are actually genetically determined, and genetic analysis is recommended in all patients with PPGL irrespective of family history or syndromic features [16,17].

PPGLs are tumors with a high genetic heterogeneity; susceptibility genes include “old genes” associated with Neurofibromatosis type 1 (NF1 gene), Von Hippel Lindau syndrome (VHL gene), and Multiple Endocrine Neoplasia type 2 syndrome (RET gene), the genes of the succinate dehydrogenase complex (SDHx genes: SDHA, SDHB, SDHC, SDHD, SDHAF2), and the more recently discovered genes TMEM, MAX, and FH. Genetic analysis is currently performed via high-throughput gene sequencing techniques (NGS: next-generation sequencing), which consent simultaneous analysis of all susceptibility genes. It is necessary to point out that NGS techniques, which detect missense, no sense, splice variants, and small intragenic insertion or deletion, do not identify large deletions and insertions. Genetic analysis must therefore include another method, such as quantitative PCR, multiple ligation-dependent probe amplification, or micro-array analyses, according to laboratory preference. Genetic testing has a direct implication to guarantee the most correct management/follow-up as pathogenic variants in different genes display some peculiar phenotypic features, and also allows cascade genetic analysis in relatives and pre-symptomatic surveillance in mutation carriers. In Table 4, major clinical phenotypes of genetic PPGL syndromes, which will be briefly described, are summarized.

### 5.1. NF1, VHL, and RET Genes

Pathogenic variants in NF1, RET, and VHL genes predispose mostly to adrenal disease, with a risk of contralateral recurrence of about 15% for NF1-related PHEO, and up to 50% for RET and VHL genes; in these patients, adrenal-sparing surgery should always be preferred.

Pheochromocytoma is a rare complication in NF1 subjects (fewer than 3% of patients) and biochemical/imaging screening is not routinely recommended, unless the patient presents signs/symptoms of catecholamines excess, so only fewer than 10% of cases reach the diagnosis thanks to biochemical testing [18]. Actually, in a recent study, the prevalence of PHEO in patients with Neurofibromatosis type 1 systematically screened by biochemical analysis and CT was up to 22.2% [19]. The mean age at diagnosis is 40 years, patients are often asymptomatic, and diagnosis is mostly incidental on computed imaging. Even if NF1-PPGL (10–12%) is rarely malignant [20], it should be appropriate, in our opinion, to consider biochemical screening at least every 24–36 months. The NF1 gene is not routinely included in PPGL-NGS panels as NF1 diagnosis is exquisitely clinical.

Pheochromocytoma develops in about 20–50% of patients with MEN2A—2B according to RET gene codon involvement [21]; in 10% of MEN2A, it is the first disease manifestation and it develops usually in the third or fourth decade of life, but in the highest-risk mutations, it can also occur at the age of eight–ten years. RET- and NF1-related PHEOs belong to molecular cluster-2, more frequently produce adrenaline, and are characterized by accessional secretion and paroxysmal symptoms.

VHL-related PHEO frequently appears at a young age (18–30 years); it is of note that VHL mutations are responsible for about 40% of paediatric PHEOs, and the tumors are mostly adrenal and frequently bilateral, even at presentation. They belong to cluster-1, and usually have elevated plasma/urinary normetanephrines with a relatively small increase in metanephrines. Extra-adrenal disease is less frequent; only rarely does PGLS develop in the head and neck compartment. Among cluster-1 PPGLs, VHL-related tumors have the lower metastatic risk, estimated to be about 3.4–8% [15,22].

### 5.2. SDHx and SDHAF2 Genes

Pathogenic variants in SDHx genes, which code for the four subunits of the succinate dehydrogenase (SDH), and the SDHAF2 gene, which code for a mitochondrial protein required for the activation of the SDH complex, are responsible for nearly 50% of hereditary PPGLs. As detailed in Table 4, pathogenic variants in these genes predispose to multiple PGLs, they differ for penetrance and inheritance model, and have a predisposition to malignant disease. SDHB gene variants together with SDHD variants account for most SDHx-related PPGLs. More than 50% of SDHB-related PGL is associated with malignant disease and SDHB mutations are responsible for 36% of all malignant PPGLs [23]. Aggressive biological behavior is also observed in SDHA- and SDHD-related PGLs [12]. Penetrance for SDHx-PGLs is generally low and is estimated to be about 25–40%. However, the SDHD gene usually predisposes to multiple non-secreting head and neck PGLs with a penetrance of 40–80% for the age of 60–70 years. Most SDHD-PGL patients may have a suggestive family history [24]. As already mentioned, hereditary predisposition to PPGL is transmitted with an autosomal dominant model; however, in the case of the SDHD and SDHAF2 genes, for the effect of maternal imprinting, the phenotype is evident mostly when the variant is transmitted by the father [25]. SDHx genes, initially thought only responsible for PPGL susceptibility, have been subsequently associated with a predisposition for kidney cancer and GIST (wild c-kit, wild PDGFRA); this must be kept in mind when setting patient follow-up.

### 5.3. TMEM and MAX Genes

The prevalence of TMEM127-MAX-related tumors is very low, respectively, less than 2% and 1% of genetic PPGL. The TMEM gene is responsible for familial PHEOs; usually, patients have a single PHEO secreting epinephrine, but bilateral involvement has been described and penetrance is probably low [26]. The MAX gene is associated with norepinephrine-producing PPGL, bilateral disease occurs in more than half of patients, and penetrance is not well defined due to the rarity of these tumors but is probably higher compared to TMEM [26].

### 5.4. FH Gene

Pathogenic variants of the fumarate hydratase gene are responsible for hereditary leiomyomatosis and papillary renal cell carcinoma (HLRCC), which is characterized by predisposition to cutaneous leiomyomata, uterine leiomyomata, and renal carcinoma. More recently, PPGLs have also been described in a few families. Penetrance for PPGLs is not known due to the small number of families; however, PPGLs can be multiple and malignant [27].

### 5.5. Other Genes

The increased availability of sequencing techniques has enabled the study of the whole exome in PPGL patients without pathogenic variants in other known susceptibility genes. So, other interesting susceptibility genes have been identified, such as EPAS1 (also called HIF2α) and EGLN1 belonging to hypoxia cluster-1 genes, while other genes are involved in mitochondrial metabolism, such as MDH2, GOT2, SLC25A11, DLST, and KIF1Bβ, and may all be ascribed to cluster-1. The genes MET and MERKT are involved in the MAP kinase pathway (cluster 2), while the H3F3A and DNMT3A genes are involved in DNA methylation. Regarding all these genes, we do not really know if they play a role in PPGL predisposition; we can argue even less about penetrance or the clinical phenotype. Some of these genes are included in commercially customed NGS panels for genetic screening in PPGL; however, clinical interpretation of pathogenic variants in these cases is not certain and communication of the result must be cautious and careful. In addition, the use of panels with a large number of genes increases the possibility of identifying gene variants with unknown significance; variants that have molecular characteristics, population frequency, and insufficient clinical and laboratory data cannot be associated with the risk of developing a disease. Genetic counseling should always be offered to the patient, especially for discussion of the genetic test result.

The identification of a pathologic variant in one of the susceptibility genes allows customizing a surveillance program according to the risk of relapse, malignancy, and other organ involvement. Cascade genetic testing should be proposed to family members only for pathogenic/likely pathogenic variants in actionable genes.

## 6. Imaging Study

Usually, imaging study by computed tomography (CT) or magnetic resonance (MRI) easily allows the identification of the PPGL location. An MRI study is probably better for parasympathetic head and neck PGLs and should be preferred for long-term follow-up in carriers of genetic variants in susceptibility genes, with the aim of reducing radiation exposure.

Conventional radiology (CT/MRI) plays an essential role in the study of non-secreting PPGLs, which are diagnosed for disorders related to a mass effect on adjacent organs or incidentally. PPGL radiological images may be variable, tumors may appear homogeneous or heterogeneous, and may have areas of hemorrhage, calcification, necrosis, or cystic evolution. In more homogeneous areas, which should be considered representative of the neoplasia, typically, unenhanced attenuation on CT is almost invariably more than 10 Hounsfield units; this feature is essential for differentiation from adrenal adenomas that have lower attenuation values. Some localizations are particularly complex to study but are typical for extra-adrenal PGLs: the head and neck, the posterior mediastinum, the Zuckerkandl body, and the sympathetic plexus of the urinary bladder, but also the heart and the kidney. In these cases, differential diagnosis with neurogenic lesion, lymphadenopathy, or sarcoma may be problematic, considering that PGLs from parasympathetic ganglia most often are not secreting.

Functional imaging constitutes another important step in the characterization of PPGL; it is essential for non-secreting PPGLs, but it is also strongly recommended in pre-operative staging because it allows the better detection of multiple locations or of the presence of metastatic disease.

Molecular signatures also impact on functional imaging; cluster-1 hypoxia-SDHx-related tumors are better studied by [68Ga]-DOTA-SSA PET/CT (Figure 1 and Figure 2), while [18F]FDG PET/CT is the second choice [28,29]; for cluster-1 hypoxia-VHL-related tumors, [18F] FDOPA PET/CT is the most sensitive, with [68Ga]-DOTA-SSA PET/CT as second choice [28,29]. For cluster-2 kinase-related tumors (NF1-RET- TMEM127, MAX), which mostly have an adrenal location, [18F] FDOPA PET/CT has a higher sensitivity and specificity due to the high uptake by the tumor compared with a normal adrenal gland, allowing for the detection of very small tumors and multiple PCCs in the same adrenal gland [28].

According to Taïeb’s review on functional imaging in the era of genomic characterization in subgroups [29], which considers all studies comparing 68Ga-DOTA-SSAs and 18F-FDOPA PET/CT, 68Ga-DOTA-SSA PET/CT might be inferior to 18F-FDOPA PET/CT in identifying small PHEOs, even if in a study on apparently sporadic PHEOs, the two methods perform equally. 68Ga-DOTATATE is also recommended in the pediatric population, due to the frequency of SDHx-related tumors in infancy.

In Table 5, the proportion of hereditary PPGLs associated with susceptibility genes, grouped by cluster, with prevalent biochemical features and suggested functional imaging, is summarized.

[123I]MIBG scintigraphy has long been routinely used for functional imaging in PPGLs, supported by studies that mainly included patients with adrenal disease; subsequent larger studies including extra-adrenal and multiple/hereditary PGL showed a sensitivity of 50–75% [30] and less than 50% for SDB-associated tumors [31]. Even if rapidly undertaken genetic diagnosis will not be complete in the initial stages of diagnostic work-up, and what has been discussed so far certainly has greater utility in the follow-up or management of metastatic disease, it can still be of help in choosing the most effective functional imaging; in patients with extra adrenal tumors, it is preferable to use [68Ga]-DOTA-SSA PET/CT, while in patients with pheochromocytoma, [18F] FDOPA PET/CT is most sensitive and accurate. However, [123I]MIBG scintigraphy has acceptable sensitivity for confirming the diagnosis of apparently sporadic PHEOs, also including non-secreting PHEOs [28] (Figure 3).

## 7. Prognostic Factors

Distinguishing PPGLS with metastatic potential from those with benign behavior is another challenge in which molecular aspects can add some useful information, in addition to anatomopathological and biological characteristics.

As previously mentioned, norepinephrine-secreting tumors, which mostly means cluster-1-associated PGL, more frequently develop malignant disease. Also, high plasmatic 3-methoxytyramine may be an expression of poorly differentiated chromaffin cells; however, SDHB deficiency is actually the most important predictor of malignancy risk [32].

The Pheochromocytoma of the Adrenal gland Scaled Score (PASS) (Table 6) is a scoring system basing on histological features [33]. PASS values ≥ 4 are indicative for metastatic risk; unfortunately, the sensitivity and specificity are low (respectively, 50% and 45%) because of the limitations associated with the subjective interpretation of the pathologist. Also, a Ki-67 index greater than 3% is a reliable indicator of tumor progression, in terms of specificity; however, in PPGL tumors, it is poorly sensible [33].

To overcome the limitations of the Pheochromocytoma of the Adrenal Scaled Score (PASS), a new scoring system, the Grading system for Adrenal Pheochromocytoma and Paraganglioma (GAPP) (Table 7), has been proposed and validated [34]. The GAPP score classifies PPGLs in well, moderately, and poorly differentiated tumors, which correlate with a 10-year survival rate of, respectively, 83%, 38%, and 0% [33].

An up-to-date GAPP score (M-GAPP), considering the secretin profile together with the loss of SDHB immunohistochemistry staining (expression of SDHB gene mutation), could be a useful tool in stratifying risk, even if it has yet to be validated.

## 8. Follow-Up

Although genetics and molecular characterization have an effective role in the early diagnostic work-up of patients with PPGL, they play an essential role in patient follow-up. PPGLs in most cases are not malignant neoplasms but tumors of uncertain behavior, patients are not evaluated by oncology specialists, and they frequently escape follow-up because it is unclear to which specialist they should be referred.

Due to the risk of local or metastatic recurrence, PPGL patients need post-surgery follow-up of at least 10 years; subjects with hereditary PPGLs, with large tumors, or other unfavorable prognostic factors need life-long surveillance [3,12,17].

Plasma and/or urinary free metanephrines should be dosed annually in the follow-up of secreting PPGLs; annual chromogranin-A assessment could be used in the follow-up of patients with pre-surgery normal levels of metanephrines and elevated chromogranin-A. Imaging will be proposed only if abnormal biochemistry is detected [33]. In subjects with a negative pre-surgery laboratory test, imaging follow-up every 1–2 years could be the better option. A surveillance program for PPGLs developed in a syndromic setting cannot be limited to the risk of recurrence of PPGL, but must consider the syndromic feature in its complexity, since the risk is also linked to other organ involvement; not secondarily, not secondarily in the same times, family members, who have tested positive for familiar mutation, must be taken into care for surveillance.

Here, we will briefly describe surveillance programs for hereditary PPGL syndrome according to different syndromic settings, which are also summarized in Table 4.

### 8.1. Neurofibromatosis Type 1

Neurofibromatosis type 1 is a hereditary neoplastic syndrome characterized by an extremely variable phenotype, with cutaneous neurofibromas and multiple café au lait macules being the most commonly known and more frequent disease manifestations. However, the clinical course of the disease is highly variable due to the involvement of other organs. NF1 patients frequently show behavioral disorders and a specific learning disability; also, deep nodular neurofibromas or voluminous plexiform neurofibromas can be responsible for neurologic deficits and severe neuropathic pain. All this leads to a worsening of the quality of life in NF1 patients at every age [33].

Neoplastic risk is related to the possible degeneration of plexiform neurofibromas into MPNST (10% of cases), and increased risk of GIST and breast cancer. All NF1 women should undergo an annual mammography, starting from the age of 25 years; in case of a positive breast cancer family history, contrast-enhanced MRI is preferred.

The prevalence of GIST varies from 3.7 to 7% in different studies [19], but no consensus exists to suggest GIST screening. Surveillance for PPGL could reasonably begin at age 14 with a dosage of plasma free or urinary fractionated metanephrines in 24 h and always in the case of pregnancy and elective surgery. For patients with a prior diagnosis of PPGL, lifelong surveillance with annual biochemical tests should be proposed [17]. If abnormal biochemistry is detected, abdominal imaging (TC- MRI) and a functional study should be performed. In NF1-mutated PPGLs, tumoral tissue shows a higher uptake of 18FDOPA compared to a normal adrenal gland, so 18FDOPA/PET/TC is the most affordable functional imaging in NF1 patients, for the detection of multiple or metastatic PPGL [12].

### 8.2. Multiple Endocrine Neoplasia Syndrome 2 (MEN2)

RET mutation carriers should be manged firstly for the risk of medullary thyroid cancer, including indication and timing for prophylactic thyroidectomy, according to American Thyroid Association (ATA) risk mutation [35]. For MTC-ATA Highest and ATA-High risk mutation carriers, prophylactic thyroidectomy is recommended in infancy. In patients with ATA-moderate risk, the time for thyroidectomy depends on the patient’s preference and family history, and annual serum calcitonin and neck ultrasound are recommended.

Surveillance for PHEO consists of a biochemical assessment from the age of 11 years for ATA HST and ATA H mutations and from the age of 16 years for ATA MOD risk mutations. If there is abnormal biochemistry, abdominal imaging is indicated (MRI-CT). As RET-mutated PHEOs belong to cluster-2, [18F]FDOPA PET/CT is considered the first choice for functional imaging; second choice is probably [68Ga]- DOTA PET/CT. Anyway, the traditional [^123/131^ I] MIBG scintigraphy is also useful [29].

### 8.3. Von Hippel Lindau Syndrome (VHL)

VHL syndrome is also characterized by multiorgan involvement; common disease complications are hemangioblastoma with cerebellar, cerebral, medullary, and retinal localization, neuroendocrine tumors, PPGL, and clear renal cell carcinoma. Surveillance with annual biochemical screening should start at the age of four as VHL mutations are often responsible of pediatric PHEOs. For renal cancer and neuroendocrine tumor detection, annual ultrasound and biennial MRI are recommended. Functional imaging should be performed only if there is a biochemical or radiological suspicion of PPGL. VHL-related PPGLs tumors behave differently from other PPGLs in clusters: they exhibit a low expression of SSTR2, and the preferable functional imaging is 18fluoro-DOPA PET/CT; otherwise, [^123/131^ I] MIBG scintigraphy and [68Ga]- DOTA PET/CT can be used.

### 8.4. Familial PPGL Syndrome (SDHx-Associated Syndromes)

Pathogenic variants in SDHx genes, initially considered as susceptibility genes only for PPGLs, are in fact also recognized to be associated with a risk of renal cancer and wild GIST (gastrointestinal stromal tumors without mutation in the KIT and PDGFRA genes). The surveillance protocol should also consider that SDHx-related PGLs can be non-secreting, so annual plasma free or urinary fractionated metanephrines must be supported by an imaging study, which should be an MRI from the base of the skull to the pelvis. After the first screening with gadolinium, follow-up should be performed with a whole-body MRI every 24–36 months. For SDHB and SDHA/C/D mutation carriers, screening should start, respectively, at the age of 6–10 and 10–15 years [12]. SDHx-mutated PPGLs strongly express the somatostatin receptor 2 (SSTR2); then, ^68^Ga-DOTATATE PET/CT is considered the most sensitive functional imaging. For SDHx mutation carriers, functional imaging should be considered at the initial screening and functional imaging is recommended if there is an abnormal finding in the biochemistry or MRI. There is no consensus on the hypothesis of alternating the MRI study with **^68^**Ga-DOTATATE PET/CT every 2–3 years.

### 8.5. TMEM- and MAX-Associated PPGL

For TMEM127 and MAX gene mutation carriers, PPGL surveillance consists of annual plasma or urinary biochemistry. Abdominal imaging can be performed every 2.3 years.

### 8.6. FH-Associated PPGL

Hereditary leiomyomatosis and papillary renal cell carcinoma (HLRCC) is a complex neoplastic syndrome characterized by strong predisposition to uterine and cutaneous leiomyomata and low penetrance for papillary kidney cancer; both FH kidney cancer and FH-PPGL may be malignant [36]. PPGLs have been described only in a small number of families; it can be argued that the penetrance is low. Mutation carriers need annual gynaecologic evaluation from the age of 20 years, and an abdominal annual MRI from the age of 8 [37]. Biochemical screening for PPGL could probably start in young adulthood (after 18 years) [38].

## 9. Conclusions

Not all pheochromocytomas–paragangliomas are the same; genomic and transcriptomic studies have taught us that the genetic characteristics of the tumor influence both the clinical presentation and the prognosis of the disease. Starting with the presentation and from the first diagnostic step, we can be guided by some clues to guarantee to the patient a diagnostic work-up that is a better fit.

Considering what we have learned, we can also in clinical practice derive some insights into classifying the type of PPGL we are dealing with as follows:-Extra-adrenal localization is most often genetically determined, so in the pre-surgery management, we need to be sure that we are not dealing with a disease with multiple localizations, considering also the thorax and head and neck compartment.-The noradrenergic phenotype is more commonly associated with an increased cardiovascular risk in terms of atherosclerotic damage, while in the adrenergic phenotype, we can more easily expect hypertensive crises, so we can intervene less empirically in preparing patients for intervention.-Extra-adrenal localization is more typical for cluster-1 PGLs, while adrenal tumors belong mostly to cluster-2, so in the choice of functional imaging, there is no doubt in preferring [68Ga]-DOTA-SSA PET/CT for extra-adrenal tumors and [18F] FDOPA PET/CT for adrenal tumors.-Extra-adrenal localization, a noradrenergic phenotype, and definite genetic alterations are associated with aggressive behavior and poor outcomes, knowing that we can, at least in part, predict the clinical course and adapt follow-up timing and inform treatment strategies.-Although the treatment of malignant PPGL is outside the topic of this article, molecular characterization reveals specific pathways and mutations that may be targeted by therapies, offering more personalized treatment options for patients.-Identifying hereditary PPGLs can assess the risk for family members and guide surveillance strategies.

Much has changed in the past two decades; in fact, the discovery of the genetic and molecular aspects of PPGLs has given a strong impetus to retrospective and prospective systematic studies that have enabled a better characterization of the disease, along with the production of consensus documents and guidelines that ensure a more rational and systemic approach to patient care.

The diffusion of this knowledge has certainly contributed to improving prognosis: consider that the surgery for PPGL, historically known to be a risky surgery with a mortality rate even higher than 40%, currently has a mortality rate of less than 1%, and this is due to earlier diagnosis, the introduction of a perioperative α-receptor blockade, and improvement in surgical techniques.

The challenge for the future is perhaps more aimed at personalizing patient follow-up, and to this point, routine, molecular characterization of PPGLs would enable us to better define the phenotype and formulate a classification based on understanding the molecular drivers; certainly, it would be desirable to perform molecular characterization on the tumor, just from the initial surgery, using NGS panels targeted to search for mutations in actionable and targetable genes. To date, the NGS study of the tumor is only performed in research clinics for malignant, already metastatic tumors; in routine surgeries, molecular characterization, when there is any, involves only immunohistochemical analysis for SDHA and SDHB proteins.

From this perspective, the next interesting task will be to see how much these new acquisitions on molecular characteristics of the disease will improve surveillance offered to patients.

In the next ten–twenty years, follow-up of health gene mutation carriers will provide useful information regarding the gene penetrance and cost-effectiveness of gene screening and instrumental surveillance for very-low-penetrance genes.

Undoubtedly, the complexity of the management from the diagnosis to the follow-up and the rarity of the disease require centralized patient management within a multidisciplinary team of experts both for patient care and for further clinical research prospects.

## Figures and Tables

**Figure 1 biomedicines-12-02385-f001:**
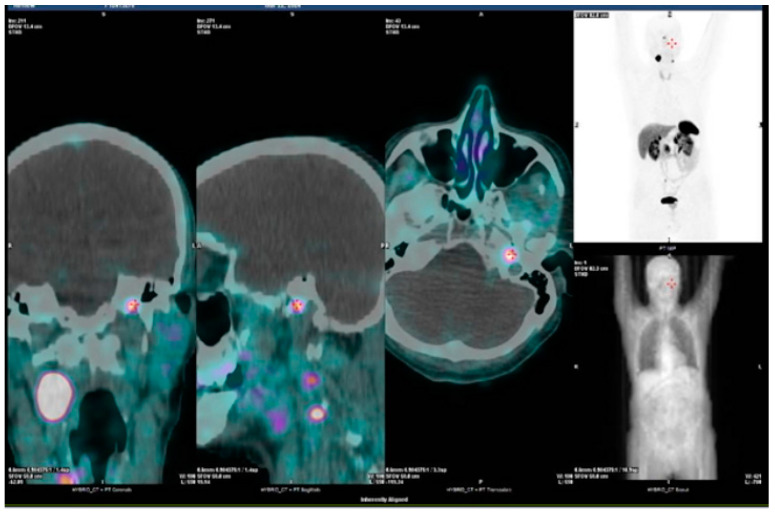
68GA-DOTA-SSA PET in SDHD-mutated patient showing bilateral HN PGLs.

**Figure 2 biomedicines-12-02385-f002:**
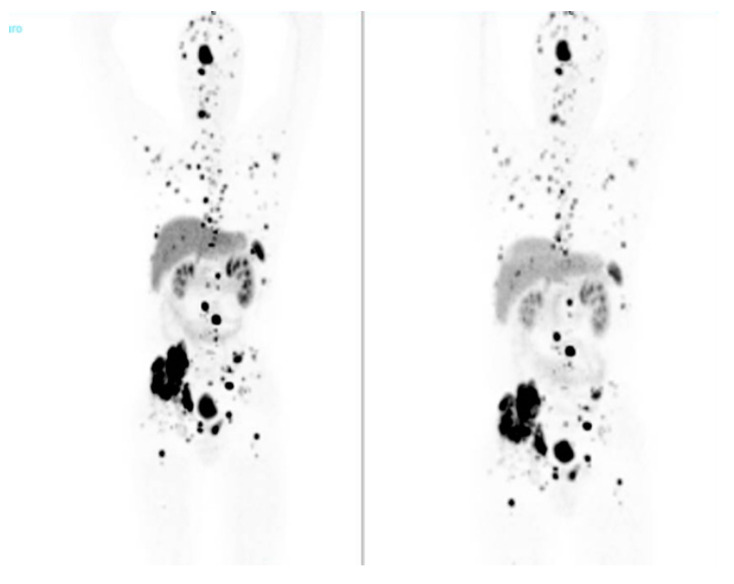
Metastatic PGL in SDHB carrier (68GA-DOTA-SSA PET).

**Figure 3 biomedicines-12-02385-f003:**
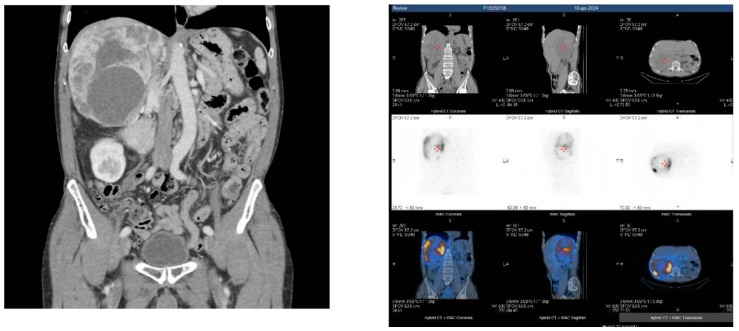
Sporadic giant right adrenal pheochromocytoma (CT scan and I123-MIBG Scintigraphy).

**Table 1 biomedicines-12-02385-t001:** Sensitivity of signs and symptoms in PPGL patients. * Significant differences between patients with or without paraganglioma, adapted from Lenders et al., 2020 [3]. ** Random effects. Paroxysmal hypertension, chest pain, flushing, and weakness were the signs/symptoms that had publication bias. Meta-analysis by Soltani et al., 2016 [4]; *** based on different definitions; meta-analysis by Soltani et al., 2016 [4]. + rare, ++ occasional, +++ frequent, ++++ very frequent

Signs and Symptoms	Frequency	Significant Differences in Patients with PHEO/PGL vs Controls *	Pooled Sensitivity (Median%; 95% CI) **
Hypertension (all)	++++	No	80.7 (74.7–85)
Hypertension (sustained)	++++		36.3 (20.5–53.9)
Hypertension (paroxysmal)	++++		36.5(24.6–49.3)
Hypertension (sustained or paroxysmal)	++++		29.4 (17.3–43.1)
Postural hypotension	++++		23–50 ***
Tachycardia or reflex bradycardia	++++	Yes	
Palpitations	++++	Yes	59.3 (51.9–66.6)
Headaches	++++	No	60.4 (53.2–67.4)
Weakness, fatigue	++++	Yes	23.8 (15.7–33.9)
Tremor	++++	Yes	20.2 (14.5–26.6)
Diaphoresis	++++	Yes	52.4 (0.46–59.1)
Classic triad (headache, diaphoresis, tachycardia)	++++		58 (28.6–84.7)
Anxiety	++++	No	28.6 (22.9–34.7)
Pallor	+++	Yes	31.6 (17.3–47.9)
Fasting hyperglycemia	++		
Nausea/vomiting	++	Yes	21.2 (16–26.7)
Weight loss	++	Yes	
Flushing	++	No	15 (9.3–21.7)
Chest pain	+		17.3 (11.4–24.2)
Abdominal pain	+		16.5 (11.9–21.6)
Paresthesia	+		13.6 (10–17.8)
Dyspnea	+		23.4 (16.2–31.5)
Dizziness	+	No	17.7 (13.5–22.3)
Decreased gastrointestinal motility	+		
Constipation	+	Yes	13.8 (32.2–29.9)
Diarrhea	+		4 (0.8–9.4)
Visual disturbances	+		9.6 (5.6–14.6)

**Table 2 biomedicines-12-02385-t002:** Indication for PPGL screening. Adapted from Lenders [3].

Clinical Contexts Requiring Screening for Pheochromocytoma and Sympathetic Paraganglioma
Patients with signs and symptoms of PPGL: spontaneous or provoked
Patients with cardiovascular events and signs/symptoms indicative for PPGL;
Finding of adrenal incidentaloma (with or without hypertension), if density is more than 10HU
Young (≤50 years), lean individuals (BMI < 25 kg/m^2^) with type 2 diabetes, with or without signs/symptoms of catecholamine excess
Carriers of a germline mutation in one of the PPGL susceptibility genes
Patients with syndromic features suggesting genetically determined or syndromic PPG
Patients with previous history or family history of PPGL

**Table 3 biomedicines-12-02385-t003:** Medications that may interfere with plasma and urinary metanephrines analysis. Adapted from Eisenhofer 2003 [11].

	Plasma		Urine	
	NMN	MN	NMN	MN
Acetaminophen	++	−	++	−
Labetalol	−	−	++	++
Sotalol	−	−	++	++
α-Methyldopa	++	−	++	−
Tricyclic antidepressants	++	−	++	−
Buspirone	−	++	−	++
Phenoxybenzamine	++	−	++	−
MAO-inhibitors	++	++	++	++
Sympathomimetics	+	+	+	+
Cocaine	++	+	++	+
Sulphasalazine	++	−	++	−
Levodopa	+	+	++	+

**Table 4 biomedicines-12-02385-t004:** Surveillance in completely resected tumor or mutation’s carriers. PGL: paraganglioma. MEN: Multiple Endocrine Neoplasia. HLRCC: hereditary leiomyomatosis and renal cell cancer. TA: thorax abdominal. HN: head and neck. Clinical evaluation: physical examination, blood pressure check. Biochemistry: plasma free normetanephrine, metanephrine, 3-methoxytyramine, and or measurement of 24 h urinary metabolite excretion. GIST: gastrointestinal stromal tumor. MPNST: malignant peripheral nerve sheet tumor. *** High risk according to PASS score and/or GAPP.

GENE (Syndrome)	CLINICAL MANIFESTATIONS									
	PPGLS								OTHER MALIGNANCIES	
	Penetrance in Carriers	Metastatic Risk	Tumor Location	Surveillance					Tumor Location	Survelliance
				Clinical Evaluation	Biochemistry	Imaging	Age of Beginning	Best Functional Imaging		
NF1 (Neurofibromatosis type 1)	3%	10–12%	Adrenal (bilateral 15%) >>TA	Every 1 year	Every 2 years	Only if abnormal biochemistry	From 14 years	^18^F-DOPA PET/CT	MPNST	Local MRI and/or ^18^FDG-PET if clinical suspicion
									GIST	Abdominal ultrasound every 2 year
									Breast cancer	Mammography every 1 year from age of 25
RET (MEN 2)	20–50%	<5%	Adrenal (bilateral 50–70%) >>>TA/HN	Every 1 year	Every 1 year	Only if abnormal biochemistry	From 8–10 years	_18_F-DOPA PET/CT	Medullary thyroid cancer	Prophylactic thyroidectomy according to MTC ATA risk
									Primary Hyperparathyrodism	Calcium metabolism every 1 year
VHL (Von Hippel Lindau Syndrome)	20–24%	5–8%	Adrenal (bilateral 50%) >>>TA/HN	Every 1 year	Every 1 year	Total body MRI every 2 years (also for other abdominal tumors)	From 4 years	^18^F-DOPA PET/CT	Central nervous system hemangiomas	MRI brain and full spine every 1–2 years from age of 16
									Retinal hemangiomas	Ophthalmic examination every year from the age of 1
									Renal cell cancer and neuroendocrine tumors	Abdominal US every 1 from the age of 12
									Endolymphatic sac tumors	Audiogram every 2–3 years from the age of 16
SDHB (PGL4)	25–40%	35–75%	TA > HN > Adrenal	Every 1 year	Every 1 year	Total body MRI every 2–3 years	From 6–10 years	^68^Ga-DOTA-SSA PET/CT	Renal cancer	Abdomen MRI every 2–3 years
									GIST	Abdomen MRI every 2–3 years
SDHD (PGL1)	40–80%	15–29%	HN > TA > Adrenal	Every 1 year	Every 1 year	Total body MRI every 2–3 years	From 10–15 years	^68^Ga-DOTA-SSA PET/CT	Renal cancer	Abdomen MRI every 2–3 years
									GIST	Abdomen MRI every 2–3 years
SDHC (PGL3)	25%	low	HN > TA > Adrenal	Every 1 year	Every 1 year	Total body MRI every 2–3 years	From 10–15 years	^68^Ga-DOTA-SSA PET/CT	Renal cancer	Abdomen MRI every 2–3 years
									GIST	Abdomen MRI every 2–3 years
SDHA (PGL6)	10%	30–66%	TA >>> Adrenal	Every 1 year	Every 1 year	Total body MRI every 2–3 years	From 10–15 years	^68^Ga-DOTA-SSA PET/CT		
SDHAF2 (PGL2)	Probably high (>50%)	low	HN >> Adrenal	Every 1 year	Every 1 year	Total body MRI every 2–3 years	From 10–15 years	^68^Ga-DOTA-SSA PET/CT		
TMEM127 (PGL5)	Unknown	low	Adrenal (bilateral 40%)>TA/HN	Every 1 year	Every 1 year	Abdomianl MRI every 2–3 years		^18^F-DOPA PET/CT		
MAX (PGL7)	Probably high	<10%	Adrenal (bilateral 50–60%) > TA/HN	Every 1 year	Every 1 year	Abdomianl MRI every 2–3 years		^18^F-DOPA PET/TCT		
FH (HLRCC)	Probably low	30%	Adrenal +TA > HN	Every 1 year	Every 1 year	Abdominal MRI	From 18 years	^68^Ga-DOTA-SSA PET/CT	Leiomyomatosis	Gynaecologic evaluation every year from age of 20
									Renal cancer	Abdominal MRI every year from age of 8
SPORADIC PPGL		PASS and/or GAPP score	TA > Adrenal	Every year for 10 years (life long if high risk * PPGL)	Every year for 10 years (life long if high risk * PPGL)	Abominal CT/US every 1 or 2 years in Adrenal location - Neck US o MRI in HN PGL		^18^F-DOPA PET/CT or 123I MIBG scintigraphy (in Adrenal location)		

**Table 5 biomedicines-12-02385-t005:** Proportion of hereditary PPGLs associated with susceptibility genes, grouped by cluster, with prevalent biochemical features and suggested functional imaging.

CLUSTER	SIGNALING PATHWAYS	GERMLINE MUTATIONS		BIOCHEMICAL PHENOTYPE	BEST FUNCTIONAL IMAGING
		GENE	Frequency		
Cluster 1 A	Pseudohypoxic Krebs cycle	SDHA	5–7%	Normetanephrine and 3methoxythyramine	68Ga-DOTA-SSA PET
		SDHB	10%		
		SDHC	1%		
		SDHD	9%		
		SDHAF2	<1%		
		FH	1%		
Cluster 1 B	Pseudohypoxia VHL/EPAS	VHL	5–7%	Normetanephrine	18F-DOPA PET/TC
		EPAS/HIF1α	Not known		
Cluster 2	Kinase signaling	RET	6%	Metanephrine	18F-DOPA PET/TC
		NF1	<2–3%		
		TMEM127	1–2%		
		MAX	1%		
Cluster 3	Wnt signaling	No germline mutations		Not known	

**Table 6 biomedicines-12-02385-t006:** PASS score: values ≥ 4 are predictive of malignancy.

PASS PARAMETERS	POINTS
Nuclear Hyperchromasia	1
Profound nuclear pleomorphism	1
Capsular invasion	1
Valscular invasion	1
Extension into periadrenal adipose tissue	2
Atypical mitotic figures	2
>3 mitotic figures/10 high-power field	2
Tumour cell spindling	2
Cellular monotony	2
High cellularity	2
Central or confluent tumour necrosis	2
Large nests or diffuse growth (>105 of tumor volume)	2
Total maximum score	20

**Table 7 biomedicines-12-02385-t007:** The tumors are classified into 3 differentiation types according to their GAPP scores—well differentiated (0–2), moderately differentiated (3–6), and poorly differentiated (7–10).

GAPP PARAMETERS	POINTS
Histological pattern	
Zellballen	0
Large and irregular nest	1
Pseudorosette (even focal)	1
Comedo-type necrosis	
Absence	0
Presence	2
Cellularity	
Low (<150 cells/U)	0
Moderate (150–250 cells/U)	1
High (>250 cells/U)	2
Ki67 labeling index (%)	
<1	0
1–3	1
>3	2
Vascular or capsular invasion	
Absence	0
Presence	2
Catecholamine type	
Non-functioning	0
Adrenergic type	0
Noradrenergic type	1
Total maximum score	10

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
