# Peer review of "Pheochromocytoma–Paraganglioma Syndrome: A Multiform Disease with Different Genotype and Phenotype Features"

_biomedicines, 2024, doi:10.3390/biomedicines12102385_

Round 1

Reviewer 1 Report

Comments and Suggestions for Authors

Dear Authors,

I have read your manuscript entitled "Pheochromocytoma-Paraganglioma syndrome: a multiform disease with different 2 genotype and phenotype features" with great interest.

I applaud the thoroughness of your review of literature on such a fascinating topic, and I found the organization of knowledge enjoyable.

I regret though that the conclusion section adds nothing to the work, reading rather coarse, and missing to leave a memorable take-home-message... I would like to invite to offer said section a second thought, and to rewrite it focusing on applicable wisdom you believe the reader should gain from your work.

I look forward to reading an updated and upgraded version of your manuscript.

Comments on the Quality of English Language

The current text often reads somewhat awkward, like at page 4, lines #93-95, when the word paroxysmal is repeated thrice in a raw adding little to the message. The are also wording imperfections, that subtract to the efficacy of the message by resulting distracting:

at line #357 the word "intreated" appears to be a typographical error.

at line #504 "can be guided from some clues" the proposition "from" should be "by"

exempli gratia.

Author Response

Dear Authors,

I have read your manuscript entitled "Pheochromocytoma-Paraganglioma syndrome: a multiform disease with different 2 genotype and phenotype features" with great interest.

I applaud the thoroughness of your review of literature on such a fascinating topic, and I found the organization of knowledge enjoyable.

Reviewer Point 1:I regret though that the conclusion section adds nothing to the work, reading rather coarse, and missing to leave a memorable take-home-message... I would like to invite to offer said section a second thought, and to rewrite it focusing on applicable wisdom you believe the reader should gain from your work.

I look forward to reading an updated and upgraded version of your manuscript.

Authors, response to point 1; Dear reviewer, thank you very much for this comment, we also agree that more organized conclusions may be more helpful to the reader. We thought of modifying the conclusions as follows, the changes will be highlighted in the test

"Not all pheochromocytomas/paragangliomas are the same, genomic and transcriptomic studies have taught us that the genetic characteristics of the tumor influence both the clinical presentation and the prognosis of the disease. Starting with the presentation and from the first diagnostic step we can be guided by some clues to guarantee to the patient the fitter diagnostic work-up.

In light of what we have learned, we can also in clinical practice derive some insights into classifying the type of PPGL we are dealing with:

  • extra-adrenal localization is most often genetically determined, so in the pre-surgery management we need to be sure that we are not dealing with a disease with multiple localizations, considering also thorax, head and neck compartment
  • noradrenergic phenotype is more commonly associated with an increased cardiovascular risk in terms of atherosclerotic damage, while in the adrenergic phenotype we can more easily expect hypertensive crises, so we can intervene less empirically in preparing patients for intervention
  • extra-adrenal localization is more typical for cluster -1 PGLs, while adrenal tumors belong mostly to cluster-2, so in the choice of functional imaging there is no doubt in preferring [68Ga]-DOTA-SSA PET/CT for extra-adrenal tumors and [18F] FDOPA PET/CT for adrenal tumors.
  • Extra-adrenal localization, noradrenergic phenotype, definite genetic alterations are associated with aggressive behaviour and poor outcomes, knowing that we can, at least in part, predict the clinical course and adapt follow-up timing and inform treatment strategies.
  • Although the treatment of malignant PPGL is outside the topic of this article, molecular characterization reveals specific pathways and mutations that may be targeted by therapies, offering more personalized treatment options for patients
  • Finally identify hereditary PPGLs can assess the risk for family members and guide surveillance strategies

Finally, molecular characterization of PPGLS would enable us to better define the phenotype and formulate a classification based on understanding the molecular drivers; certainly, it would be desirable to perform molecular characterization on the tumor, right from the initial surgery using NGS panels targeted to search for mutations in actionable and targetable genes. To date, NGS study of the tumor is done only in Research Clinics on malignant, already metastatic tumors; in routine surgeries, molecular characterization, when there is any, involves only immunohistochemical study for SDHA and SDHB proteins

Reviewer Point 2 : Comments on the Quality of English Language

The current text often reads somewhat awkward, like at page 4, lines #93-95, when the word paroxysmal is repeated thrice in a raw adding little to the message. The are also wording imperfections, that subtract to the efficacy of the message by resulting distracting:

at line #357 the word "intreated" appears to be a typographical error

at line #504 "can be guided from some clues" the proposition "from" should be "by"

exempli gratia.

Authors, response to point 2; Dear reviewer thanks for your suggestion,

page 4, lines #93-95: we have tried to rephrase it avoiding unnecessary repetition:

"However, the appearance of episodic symptoms, the so called “spells” always constitutes a valid diagnostic clue: some patients experience symptoms in a paroxysmal manner, sometimes associated with increase in blood pressure values: the frequency of attacks is extremely variable, ranging from a few attacks over several years to more than one per day"

at line #357 the word "intreated": we have corrected with:  Even if rapidly undertaken genetic diagnosis……

at line #504: thanks, we have corrected

Reviewer 2 Report

Comments and Suggestions for Authors

The manuscript provides a thorough review of Pheochromocytoma and Paraganglioma (PPGL), focusing on the genetic and clinical aspects of the disease. It highlights the importance of genetic screening, surveillance protocols, and the implications of various mutations associated with PPGL. The review is well-structured and offers valuable insights into the management and understanding of this complex syndrome.

Comments:

1-     While the review covers a broad range of topics, it would be beneficial to include more recent studies or meta-analyses that provide updated statistics or findings related to PPGL. This would enhance the manuscript's relevance and provide a more comprehensive overview of the current state of research.

2-     The discussion on the clinical implications of genetic findings is crucial. However, the manuscript could elaborate more on how these findings influence treatment decisions and patient management strategies. Including case studies or examples could strengthen this section.

3-     Consider adding more figures or tables to summarize key data, such as mutation prevalence or clinical outcomes associated with different genetic profiles. Visual aids can enhance understanding and retention of complex information.

4-     The section on imaging studies is informative but could be expanded to include a comparison of the sensitivity and specificity of different imaging modalities. A table summarizing this information would be beneficial.

5-     The conclusion summarizes the key points effectively but could be expanded to include future directions for research in PPGL. Discussing potential areas for further investigation would provide a forward-looking perspective

Author Response

Comments and Suggestions for Authors

The manuscript provides a thorough review of Pheochromocytoma and Paraganglioma (PPGL), focusing on the genetic and clinical aspects of the disease. It highlights the importance of genetic screening, surveillance protocols, and the implications of various mutations associated with PPGL. The review is well-structured and offers valuable insights into the management and understanding of this complex syndrome.

Comments:

Reviewer Point 1 -  While the review covers a broad range of topics, it would be beneficial to include more recent studies or meta-analyses that provide updated statistics or findings related to PPGL. This would enhance the manuscript's relevance and provide a more comprehensive overview of the current state of research.

Authors, response to point 1 - Dear reviewer thank you for your comment. Following you suggestion we looked for more recent studies or meta-analyses that provide updated statistics or findings related to PPGL.

We find an interesting meta-analysis by Cuspidi regarding Cardiac involvement. We have included it in the article: the changes will be highlighted in the test

“a recent meta-analysis by Cuspidi et al (8), suggest that very early abnormality in LV systolic function not detectable by conventional echocardiography in PPGLs patients may be detected by speckle tracking echocardiography, which investigate left ventricular mechanics via global longitudinal strain, a more sensitive index of left ventricular function compared to ejection fraction. This means that probably cardiac involvement in much more frequently than previously believed”

We also put more emphasis on the meta-analysis by Crona et al “Genotype-phenotype correlations in pheochromocytoma and paraganglioma: a systematic review and individual patient meta-analysis.” Endocrine-related cancer vol. 26,5 (2019): 539-550. doi:10.1530/ERC-19-0024, that we had already mentioned in the article. the changes will be highlighted in the test

“An interesting meta-analysis by Crona et al (15) on 703 patient from 21 studies, aimed to evaluate if driver mutation are valuable to predict survival and malignant disease, has classified patients according to different molecular subgroups according to SDHB mutation, attribution to cluster-1 or 2 or 3, and to four subgroups (tricarboxylic acid cycle, VHL/EPAS1, kinase signalling or Wnt/unknown): the meta-analysis concluded that in multivariate model only SDHB mutation, norepinephrine and dopamine phenotype are correlated with the risk of metastasis, while no association is found for other molecular systems”

Other meta-analyses are less interesting with regard to our topic and concern 1) pre-operative use of selective vs non-selective alpha-blockade in PPGLs patients undergoing surgery Yadav, Sanjay K et al. Indian journal of endocrinology and metabolism vol. 26,1 (2022): 4-12. doi:10.4103/ijem.ijem_469_21; 2) effect of chemotherapy with CVD on tumour volume in patients with malignant PPGL, Niemeijer, N D et al Clinical endocrinology vol. 81,5 (2014): 642-51. doi:10.1111/cen.12542, 3) surgery strategy Schiavone, Donatella et al. “Total adrenalectomy versus subtotal adrenalectomy for bilateral pheochromocytoma: meta-analysis.” BJS open vol. 7,6 (2023): zrad109. doi:10.1093/bjsopen/zrad109.

Reviewer Point 2 The discussion on the clinical implications of genetic findings is crucial. However, the manuscript could elaborate more on how these findings influence treatment decisions and patient management strategies. Including case studies or examples could strengthen this section.

Authors, response to point 2; Thank you also for this comment, we thought we would make the discussion more practical by identifying take-home messages that could serve as reference points for clinical practice. We have extended the discussion as follows, the changes will be highlighted in the test:

“Not all pheochromocytomas/paragangliomas are the same, genomic and transcriptomic studies have taught us that the genetic characteristics of the tumor influence both the clinical presentation and the prognosis of the disease. Starting with the presentation and from the first diagnostic step we can be guided by some clues to guarantee to the patient the fitter diagnostic work-up.

Considering what we have learned, we can also in clinical practice derive some insights into classifying the type of PPGL we are dealing with:

  • extra-adrenal localization is most often genetically determined, so in the pre-surgery management we need to be sure that we are not dealing with a disease with multiple localizations, considering also thorax and head and neck compartment
  • noradrenergic phenotype is more commonly associated with an increased cardiovascular risk in terms of atherosclerotic damage, while in the adrenergic phenotype we can more easily expect hypertensive crises, so we can intervene less empirically in preparing patients for intervention
  • extra-adrenal localization is more typical for cluster -1 PGLs, while adrenal tumors belong mostly to cluster-2, so in the choice of functional imaging there is no doubt in preferring [68Ga]-DOTA-SSA PET/CT for extra-adrenal tumors and [18F] FDOPA PET/CT for adrenal tumors.
  • extra-adrenal localization, noradrenergic phenotype, definite genetic alterations are associated with aggressive behaviour and poor outcomes, knowing that we can, at least in part, predict the clinical course and adapt follow-up timing and inform treatment strategies.
  • although the treatment of malignant PPGL is outside the topic of this article, molecular characterization reveals specific pathways and mutations that may be targeted by therapies, offering more personalized treatment options for patients
  • identify hereditary PPGLs can assess the risk for family members and guide surveillance strategies”

Reviewer Point 3  Consider adding more figures or tables to summarize key data, such as mutation prevalence or clinical outcomes associated with different genetic profiles. Visual aids can enhance understanding and retention of complex information.

Reviewer Point 4 The section on imaging studies is informative but could be expanded to include a comparison of the sensitivity and specificity of different imaging modalities. A table summarizing this information would be beneficial

Authors, response to point 3 and 4;

Thank you also for this suggestion, in table 4 are already summarized PPGLs location, syndromic features and risk of malignancy, hence we have add a new table (table 5) with the three molecular clusters, mutation prevalence, and suggestion for functional imaging. Since the comparison in terms of sensitivity and specificity between the various methods is not well defined and would need more extensive discussion, which is probably excessive for the structure of our article, we have added and mentioned a paragraph referring to TaÑ—eb's review and mentioning the comparison between the 68Ga-DOTA-SSAs and 18F-FDOPA PET/CT. We have included in the article, the changes will be highlighted in the test

“According to TaÑ—eb's review on functional imaging in the era of genomic characterization in subgroups (31), which consider all studies comparing 68Ga-DOTA-SSAs and 18F-FDOPA PET/CT, 68Ga-DOTA-SSA PET/CT might be inferior to 18F-FDOPA PET/CT to identify small PHEOs, even if in a study on apparently sporadic PHEOs, the two methods perform equally. 68Ga-DOTATATE is also recommended in the pediatric population, due to frequence of SDHx related tumors in infancy.

In table 5  proportion of hereditary PPGL associated to susceptibility genes, grouped by cluster, with prevalent biochemical features and suggested functional imaging, are resumed

Reviewer Point 5 The conclusion summarizes the key points effectively but could be expanded to include future directions for research in PPGL. Discussing potential areas for further investigation would provide a forward-looking perspective.

Authors, response to point 5: Thanks, it is a good suggestion to conclude with a perspective that considers new research insights and constructions of pathways for improvement in clinical practice.

“Much has changed in the past two decades, in fact the discovery of the genetic and molecular aspects of PPGLs has given a strong impetus to retrospective and prospective systematic studies that have enabled a better characterization of the disease, the production of consensus documents and guidelines that ensure a more rational and systemic approach to patient care. The diffusion of this knowledge has certainly contributed to improving prognosis: consider that the surgery for PPGL, historically known to be a risky surgery with a mortality rate even higher than 40%, has currently a mortality rate of less than 1%, and this is due to earlier diagnosis, introduction of perioperative α-receptor blockade and improvement in surgical techniques. The challenge for the future is perhaps more aimed at personalizing patient follow-up, and to this point routine, molecular characterization of PPGLs would enable us to better define the phenotype and formulate a classification based on understanding the molecular drivers; certainly, it would be desirable to perform molecular characterization on the tumor, just from the initial surgery using NGS panels targeted to search for mutations in actionable and targetable genes. To date, NGS study of the tumor is done only in Research Clinics for malignant, already metastatic tumors; in routine surgeries, molecular characterization, when there is any, involves only immunohistochemical analysis for SDHA and SDHB proteins.

In this perspective the next interesting task will be to see how much these new acquisitions on molecular characteristics of the disease will improve surveillance offered to patients. In the next ten-twenty years follow-up of health gene mutation carriers will provide useful information regarding gene-penetrance and cost-effectiveness of gene screening and instrumental surveillance for very low penetrance genes

Undoubtedly, the complexity of the management from the diagnosis to the follow-up and the rarity of the disease require centralised patient management within a multidisciplinary team of experts both for patient care and for further clinical research prospects”

Reviewer 3 Report

Comments and Suggestions for Authors

This is a very detailed narrative review about pheochromocyroma/

paraganglioma with a lot of focus on new genetic progresses and their impact on management guidelines.

Given the detailed nature of the manuscript (extremely detailed in the part of genetics, molecular biology, specific monitoring and follow-up guided by these aspects) there is a relative lack of adequate discussion of the challenging diagnosis of non-functional paragangliomas in different localisations. These clinical aspects (if the manuscript is intended to be so broad) are to be discussed or the title should be changed to illustrate that only functional tumors are addressed.

Author Response

Comments and Suggestions for Authors

This is a very detailed narrative review about pheochromocytoma/paraganglioma with a lot of focus on new genetic progresses and their impact on management guidelines.

Reviewer Point 1 Given the detailed nature of the manuscript (extremely detailed in the part of genetics, molecular biology, specific monitoring and follow-up guided by these aspects) there is a relative lack of adequate discussion of the challenging diagnosis of non-functional paragangliomas in different localisations. These clinical aspects (if the manuscript is intended to be so broad) are to be discussed or the title should be changed to illustrate that only functional tumors are addressed.

Authors, response: Dear reviewer, thank you very much for your suggestion, we added a paragraph with more details on radiological characterization related to non-secreting forms and abnormal localization. The changes will be highlighted in the test

“Conventional radiology (CT/MRI) plays an essential role in the study of non-secreting PPGLs, which are diagnosed for disorders related to a mass effect on adjacent organs or incidentally. PPGLs radiological images may be variable, tumors may appear homogeneous or heterogeneous, may have areas of hemorrhage, calcification, necrosis or cystic evolution. In more homogeneous area, which should be considered representative of the neoplasia, typically unenhanced attenuation on CT is almost invariably more than 10 Hounsfield units, this feature is essential for differentiation from adrenal adenomas that have lower attenuation values. Some localizations are particularly complex to study, but are typical for extradrenal PGLs: the head and neck, the posterior mediastinum, the Zuckerkandl body the sympathetic plexus of the urinary bladder, but also the heart and the kidney: in these cases differential diagnosis with neurogenic lesion, lymphadenopathy or sarcoma may be problematic, considering that PGL from parasympathetic ganglia most often are not secreting.

Functional imaging constitutes another important step in the characterisation of PPGL,it is essential for non-secreting PPGLS but it is also strongly recommended in pre-operative staging because it allows better detection of multiple locations or of the presence of metastatic disease”.

Round 2

Reviewer 3 Report

Comments and Suggestions for Authors

The manuscript is improved according to previous suggestions, the actual version is clear, detailed and provides a useful overview on this rare pathology. I have no further issues to raise